# Attachable Inertial Device with Machine Learning toward Head Posture Monitoring in Attention Assessment

**DOI:** 10.3390/mi13122212

**Published:** 2022-12-14

**Authors:** Ying Peng, Chao He, Hongcheng Xu

**Affiliations:** 1Normal College of Liupanshui, Liupanshui 553000, China; 2School of Mechano-Electronic Engineering, Xidian University, Xi’an 710071, China

**Keywords:** attachable device, head posture, attention, machine learning, inertial measurement

## Abstract

The monitoring of head posture is crucial for interactive learning, in order to build feedback with learners’ attention, especially in the explosion of digital teaching that occurred during the current COVID-19 pandemic. However, conventional monitoring based on computer vision remains a great challenge in the multi-freedom estimation of head posture, owing to low-angle annotation and limited training accuracy. Here, we report a fully integrated attachable inertial device (AID) that comfortably monitors in situ head posture at the neck, and provides a machine learning-based assessment of attention. The device consists of a stretchable inertial sensing unit and a fully integrated circuit-based system, as well as mechanically compliant encapsulation. Due to the mechanical flexibility, the device can be seamlessly attach to a human neck’s epidermis without frequent user interactions, and wirelessly supports six-axial inertial measurements, thereby obtaining multidimensional tracking of individual posture. These head postures (40 types) are then divided into 10 rotation actions which correspond to diverse situations that usually occur in daily activities of teaching. Benefiting from a 2D convolutional neural network (CNN)-based machine learning model, their classification and prediction of head postures can be used to analyze and infer attention behavior. The results show that the proposed 2D CNN-based machine learning method can effectively distinguish the head motion posture, with a high accuracy of 98.00%, and three actual postures were successfully verified and evaluated in a predefined attention model. The inertial monitoring and attention evaluation based on attachable devices and machine learning will have potential in terms of learning feedback and planning for learners.

## 1. Introduction

Online learning became an alternative candidate to conventional teaching during the current COVID-19 pandemic, and head posture is a key index that is highly relative to learners’ attention [1,2,3,4,5,6,7]. Continuous monitoring of head motions, therefore, can build direct recognition and feedback of learning attention, in order to assist the learner in gaining high-efficiency and comfortable experiences during long-term and boring teaching. Currently, three primary approaches have been used to track head posture. The first option applies facial feature recognition based on deep learning to infer head posture in attention assessment. However, changes in both learner and learning scenes limit the feature extraction of face images, which only establishes indirect relations between images and attention [8,9]. Another alternative route is to use an eyeball instrument to collect the learner’s eyeball dynamic data for long-term analysis of head posture; these methods generally require sophisticated, expensive instruments, and can even disturb the learning status of the subject [10,11]. In addition, the learner’s head posture can be obtained through continuous image streams. However, the image quality is sensitive to changes in the angle and position of camera installation, resulting in low-angle annotation and limited training accuracy [12,13,14,15,16]. Therefore, it is urgently necessary to obtain reliable analysis of attention, in order to achieve continuous and robust monitoring of head posture. 

Up to now, flexible electronics have provided an inspired choice for continuous motion monitoring [17,18,19,20]. Particularly during the monitoring of head posture, a portable inertial device can support noninvasive, long-term, and wireless in-situ measurement in tracking weak movements and rotations, which can build an intimate relationship between the posture feature and inertial motion. For instance, Liu et al. developed real-time and online inertial tools and devices, the wearable Human Activity Recognition (HAR) system [21,22], HAR Research Pipeline [23], and ASK2.0 [24], for continuous smart monitoring and recognition of various human behaviors. Simultaneously, convolutional neural network (CNN)-based methods have successfully been used to recognize and classify various human signs [25,26,27]. Therefore, combining the inertial monitoring of wearable devices with feature clustering based on CNN-based machine learning can provide a convenient and reliable judgment of attention assessment [28,29,30,31]. Compared with an estimation of the head’s image stream, this method is without limits for variations in the posture sample and ambient light, as well as for feature extraction. The posture feature is only obtained through inertial movement and rotation. For example, Mao et al. used 3-axial angular velocity to measure head posture, and established seven states-based evaluations for the learner’s attention model. Since the measured result is low-dimensional and sparse, the designed model is difficult to generalize evaluation with [32]. Brandt et al. applied head movements to drive a wearable camera, and established a direct relation between the image stream and head movements. However, estimation of the head posture with visual images limits extraction features, consequently leading to high analyzing error [33]. In addition, Hua [13], Zhang [34], Liu [35,36], and Xu et al. [37] proposed a computer vision-based method for head posture monitoring; their tested sample merely relied on manual collection at the lab, which may be too sparse and difficult to meet practical needs. 

To overcome these obstacles, we developed a fully integrated attachable device that supports the in-situ 6-axial inertial measurement of head postures at the neck. The attachable inertial device (AID) integrates MCU, Bluetooth, and Li^+^ battery units, with a gyroscope and acceleration sensor in a flexible patch can track the rotation and axial movement of head postures. Due to its mechanical flexibility, the device can be seamlessly attached to the human neck’s epidermis without frequent user interactions, and wirelessly supports six-axial inertial measurement, thereby obtaining multidimensional tracking of individual posture. These head postures (40 types) were divided into 10 actions that corresponded to diverse situations that usually occur during the daily activities of teaching. A 2D CNN-based machine learning method classified and predicted different head postures, and was able to analyze and infer corresponding attention behaviors. The results showed that the 2D CNN-based machine learning model could effectively and accurately distinguish learners’ head motion postures, with a high accuracy of 98.00%, and three actual postures were successfully verified and evaluated in a predefined attention model. We expect that this design and validation of a fully integrated attachable device with machine learning paves the way for powerful means of attention assessment in learning feedback and planning. 

## 2. Method

### 2.1. Design of the Circuit Schematic Diagram

The inertial unit (MPU6050, INVENSENSE) was connected with an 8-bit MCU chip (Atemega328p, ATMEL) via the I2C communication protocol. The Bluetooth module (cc2640, RF-STAR, China) was communicated via a serial protocol to the processing center. The working frequency of the crystal oscillator in the circuit was 8 MHz, in order to decrease the power consumption, and the entire circuit was powered by a Li^+^ polymer battery with a voltage of 3.2~3.7 V (Dajia Manyi Technology Co., Ltd., Shenzhen, China).

### 2.2. Fabrication of the Attachable Inertial Device

An initial planar circuit board based on polyimide was fabricated at Shenzhen Muwei Electronic Co., Ltd. (Shenzhen, China). The outline of the stretchable circuit was engraved with a 355 nm UV laser (50 kHz pulse frequency, 300 mm s^−1^ cutting speed for 5 repetitions of cutting, YLCF65UV, Yuanlu Photoelectric Technology, Wuhan, China). All of the electronic components and the battery were then soldered in a soldering box at 200 °C for 10 min. Next, the predesign mold was input to a CNC engraving machine (3040, Mikie CNC) with a cutting speed of 22,000 mm min^−1^, in order to cut an Al plate with a thickness of 5 mm. Colored Eco-flex elastomer (Smooth-on, Macungie, PA, USA) was poured into the mold to form the bottom packaging with a thickness of 0.8 mm; it was cured at ambient temperature for 30 min. Furthermore, the soldered circuit was aligned at the mold, and the remaining elastomer was poured into the mold to cover the circuit which was placed in a vacuum for 20 min to eliminate air. Finally, a glass plate was used to press on the uncured Eco-flex to form the resulting device, which was cured at ambient temperature for 30 min. 

### 2.3. Tested Process of Head Postures

The inertial results of head posture were based on continuous motional posture as participants (2 adult males and 1 adult female (age: 22~30 years old, height: 160~190 cm, weight: 45~100 kg)) performed special movements: rightward rotation (θ_Yaw_ = 10~85°), bow (θ_Pitch_ = 10~80°), and roll (θ_Roll_ = 10~50°). There were 40 movement types in total, which corresponded to diverse situations that usually occur during daily activities of teaching. Each procedure at every angle was performed 1000 times repeatedly, and the tested angle interval was 5°. In addition, to improve the robustness of the training model, the test process was added with some special motions, such as leftward or frontal deflection at the same time, in order to simulate head motions that frequently occur in daily activities; the frequency of these movements is around 20.00%. The participants signed an ethical informed consent in the form of a Consent Form file in the Appendix A.

## 3. Results and Discussion

### 3.1. Design of the Inertial Device 

Long-term measurements of head posture at the human neck require a device that is comfortable, skin-attachable, wireless, and noninvasive. Figure 1a depicts the electrical and structural illustration of the fully integrated device for monitoring head posture. It primarily consists of an integrated MCU, Bluetooth, Li^+^ polymer battery, and an inertial unit fed with a stretchable connection (Appendix A); it was encapsulated in an ultra-elastic shell (ε = 60 kPa). The ultra-elastic encapsulation matched the stiffness of the skin epidermis, in order to make the wearer comfortable. Figure 1b is the optical diagram of the fabricated device mounted on a subject’s neck, and the insert exhibits its small size near a coin. Benefiting from its integration and flexibility, the device can easily obtain continuous neck motions without frequent user interactions. Based on the 6-axial inertial data from head postures, the CNN method can be used to train and predict head movement states, thus providing verification in attention assessment (Figure 1c). The flowchart of the complete system from data acquisition and processing, to posture training and prediction, finally to real-posture verifications, is shown in Appendix A. As shown in Figure 1d, the inertial unit is mainly composed of a gyroscope and an acceleration sensor (Appendix A) that are connected with processing, controlling, and transmitting units. Figure 1e shows that the stretchable device can withstand 20% tensional strain under uniaxial stretching, and 45° bending during finite element analysis (FEA); these conditions are necessary for epidermal deformations during physical wearing. Furthermore, as also demonstrated in Figure 1f, the fabricated device can bear severe stretching (20%), twisting (90°), and bending (45°), showing its potential for intimate skin attachability. 

Figure 2a shows the hybrid fabricated process of the fully integrated AID. It primarily consists of three key steps: (1) stretchable outline of the bare circuit engraved by the UV laser; (2) reflow soldering at low temperature; and (3) full encapsulation of the bare circuit into the ultra-elastic silicone. Details of the fabricated configuration are shown in the Methods section. To observe the influence of encapsulation on signal quality, different thicknesses of the eco-flex were used for encapsulation of the bare circuit. Figure 2b depicts that no observable attenuation of the signal intensity occurred at the inertial units, and the Bluetooth’s signal amplitude decreased with an increase in thickness. This could possibly contribute to attenuation of the electromagnetic wave across a thick obstruction, but the signal intensity remained a stable wireless transmission at 2.4 GHz. To further promote attachability close to the skin’s elasticity, ultra-elastic silicone was compared with other general elastomers (Figure 2c). The result shows that the colored silicone exhibited superior elasticity (~60 kPa) near the skin epidermis (~20 kPa), thereby making the device soft and attachable for long-term wearing on skin. 

### 3.2. Inertial Measurements of Head Posture 

Figure 3a illustrates neck-posture images tested with the AID, and the corresponding results. It can be observed that neck rotation degree is almost proportional to the amplitude of the inertial motion around the corresponding rotating axis. Moreover, each rotation consists of several inconsistent peaks and different waves where the frequency spectrum could distinguish every weak vibration. Figure 3b also exhibits similar proportional relations between the rotating angle and inertial response. Therefore, the head posture’s feature can be clearly obtained based on the output amplitude of inertial movements and rotations along the yaw and pitch directions. Figure 3c shows the wearable device’s lowest resolution in both measurements of the acceleration and angular velocity, with the pitch angle of the tested posture at 5°, and their corresponding resolutions were 0.25 g and 0.4 red/s, respectively, showing its superior sensitivity in monitoring inertial motion. Figure 3d,e are the frequency responses for the amplitude and power spectrum as the tested feature was observed via Fourier transform (FT), respectively. The motion response primarily gathers around 0.87 Hz, which was consistent with physical head movement. Other frequency peaks may be attributed to weak vibrations of head coupling with the wearable device. In addition, for the special posture as the subject performs a rightward deflection of θ_Yaw_ = 80° (the coordinate direction is defined as shown in Figure 1b), the inertial results of the posture around their its corresponding rotating axes are shown in Figure 3f via short-time FT, which further exhibits the long-term measured stability of the wearable device. Furthermore, the frequency feature is almost identical with the above FT’s result. Owing to the skeletal complexity of the neck, each movement produces abundant offsets and vibrations, thereby resulting in rotation and movement around a single minor direction. It was verified, as shown in Figure 3g, that the head motion is not an individual rotation; instead, it is a multidimensional motion in space, from which it can be inferred that the computer vision-based correlative model had an obvious sample estimation error, and was without sufficient accuracy during training. 

### 3.3. Machine Learning in Head Posture Estimation

According to the above testing, the result of an individual head posture consists of six-dimensional inertial data. To extract these 1D features, the convolutional neural network randomly reduced the dimension of the six-axial data to simulate two-dimensional feature extraction, without other pre-setting, in order to achieve end-to-end training, thereby ensuring accuracy and credibility. The multiparameter 2D CNN-based machine learning method was built to be divided into multi-independent blocks, as shown in Figure 4a, and it corresponds to the pseudocode template shown in Appendix A. All posture features of different positions and deflections were initially normalized and read, and the feature length and dimension were 1000 and 6, respectively. Then, the data matrix was reduced at dimension by the Triplet function (*L_tr_*) to the max-pooling layer after the convolution and nonlinearization. The dropout function prevented features from overfitting, and the batch normal function normalized features. Furthermore, processed features were classified into 10 classes through the Cross-entropy function (*L_ce_*). Finally, the features of different actions at the full-connecting layer were predicted by the one-hot function. Each block underwent multiple convolutions and pooling, thereby lowering the convergence time during training.

The t-Distributed Stochastic Neighbor Embedding (t-SNE) method clustered all features to visualize, and the confusion matrix plotted the prediction rate, as shown in Figure 4b, from the initial to the 40th iterations. It is obvious that feature overlapping was high early in training, thus resulting in low predicted accuracy. With further iterations and training, features became clustered with a superior accuracy of 86.80% until the 20th iteration. This is because each independent block consisted of multiple convolutions and pooling that could repeatedly extract and classify inner goals. Finally, all features became totally divided as the training reached the 40th iteration, with a predicted accuracy of 94.50%, indicating fast convergence ability of the design of independent blocks for feature extraction and classification. 

### 3.4. Applications for Attention Assessment 

As shown in Figure 5a, the predicted feature matched the training with an accuracy of 98.00%. However, some rolling actions remained with low convergence accuracy, which may have been caused by the following: (1) low iterations during training; (2) insufficient samples of head postures; (3) the subject made nonstandard rolling postures. Moreover, the roll change was not directly related to the head’s attention. The effect of the roll change on the training model could, therefore, be ignored. The trained accuracy exceeded 90.00% after 10 iterations, and the tested accuracy was greater than 80.00% after 20 iterations, as shown in Figure 5b, which is highly consistent with the t-SNE result, further exhibiting the model’s fast convergence ability. Furthermore, both normalized losses of the object functions, *L_tr_* and *L_ce_*, were less than 0.1 prior to 20 iterations (Figure 5c), showing its clustering ability. The trained model also tested other head postures of three subjects who were not tested in the above training, as shown in Figure 5d. When three subjects performed the special postures rightward rotation (θ_Y_ = 40°), slight bow (θ_p_ = 10°), and frontal rolling (θ_R_ = 30°), their corresponding inertial results were input to the independent block model driven by the 2D CNN-based machine learning method. The tested accuracy could be used to infer the corresponding head posture. As shown in Figure 5e, the tested accuracies of the three rotations are highly consistent with that of the training, with a rate over 80.00%; moreover, the tested posture corresponded to the trained accuracy of the above posture angle. These results show that the independent block in machine learning can accurately distinguish the learner’s head posture. Furthermore, head postures were defined as the corresponding normal and alert states during learning, as shown in Figure 5f. Only with a yaw angle over 25° and a pitch angle over 15° could the head attention can be expressed as an “Alert!” state; the other postures are expressed as normal states. Therefore, the three tested postures corresponded to learning attentions that could be depicted as inattentive, attentive, and irrelevant, respectively. In addition, compared with several other methods [16,21,25,38,39,40,41,42] for human activity recognition, our proposed model exhibited superior abilities, as shown in Appendix A, including wearability and wireless acquisition for data, as well as smart recognition. Such wireless attachable patch-based machine learning tracking could potentially pave a way for wearable monitoring and evaluation. 

## 4. Conclusions 

In this study, we developed an attachable device that consists of a stretchable inertial sensing unit and a fully integrated circuit-based system to monitor in situ head postures, and provides a machine learning-based assessment of attention. Due to its mechanical flexibility, the device can be seamlessly attached to the human neck epidermis without frequent user interactions. In addition, the inertial unit is composed of a gyroscope and an acceleration sensor with six-axial measurement ability, thereby supporting a continuous multidimensional monitoring for each individual head posture. These head postures (40 types) are then divided into 10 rotation actions which correspond to diverse situations that usually occur during daily activities of teaching, and are further classified and predicted with a two-dimensional convolutional neural network-based machine learning model. The trained and predicted results can effectively distinguish head motion posture with a high accuracy of 98.00%, and the actual posture was successfully verified and evaluated in a predefined attention model. The inertial monitoring and attention evaluation based on both attachable devices and machine learning has the potential to provide learning feedback and planning for learners.

## Figures and Tables

**Figure 1 micromachines-13-02212-f001:**
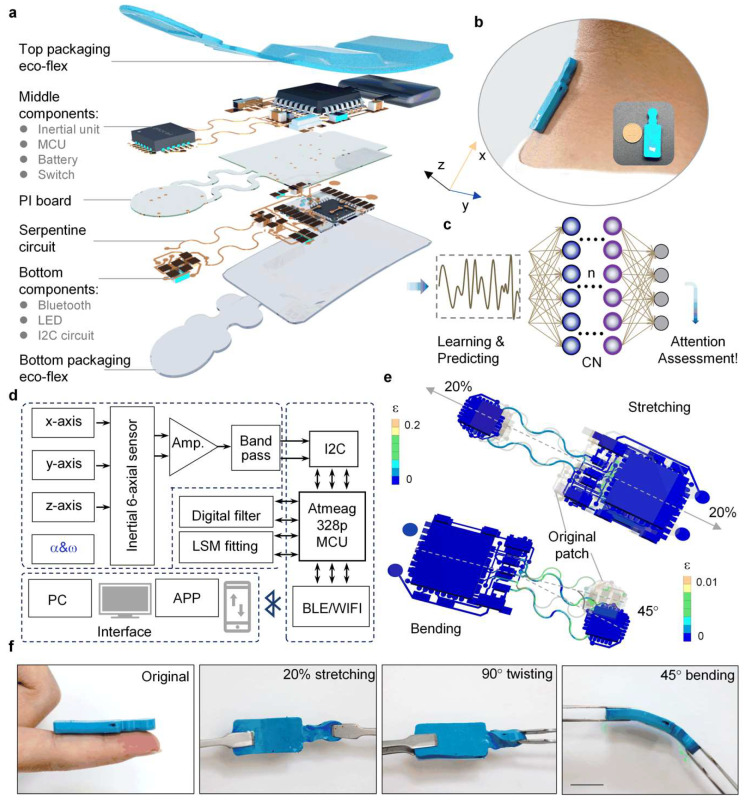
Design and mechanical deforming ability of the AID. (**a**) Mechanical structural and electrical schematic diagram of the AID. (**b**) Optical image of AID attached to the neck’s epidermis, and a corresponding frontal photo compared with a coin. (**c**) Attention assessment process via machine learning based on the monitoring of head postures. (**d**) Electrical block diagram of inertial signal processing and transmitting. (**e**) FEA’s strain diagram of the AID under 20% stretching and 45° bending deformations. (**f**) Images of the AID in elastic compliance in no load, torsion, twisting, and bending. Scale bar, 1 cm.

**Figure 2 micromachines-13-02212-f002:**
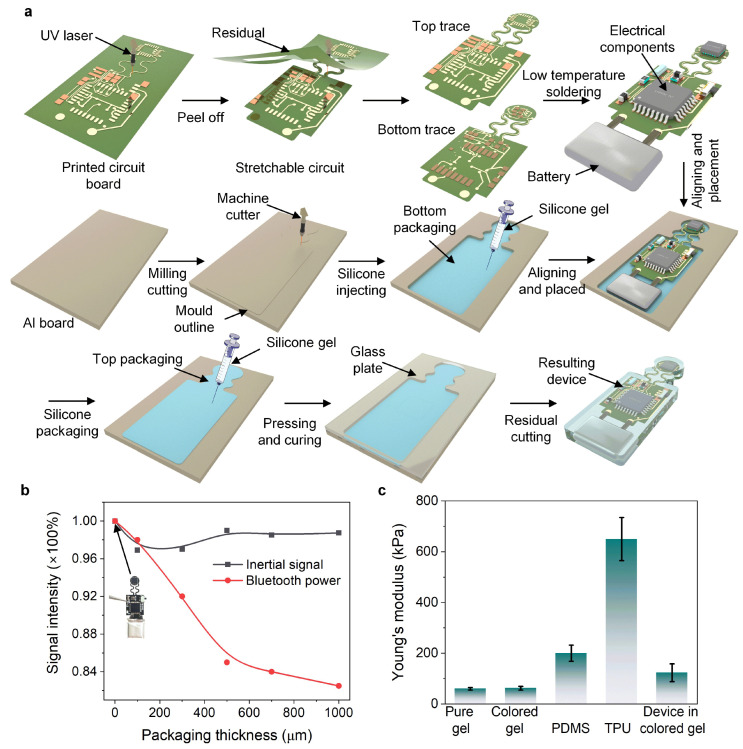
Fabrication of the AID. (**a**) Laser engraving of the stretchable bare circuit and the encapsulation. (**b**) Variations in the signal intensity of the inertial signal and Bluetooth output with varying thicknesses of packaging silicone gel. (**c**) Tested Young’s moduli of different elastomers and the AID encapsulated by the colored silicone.

**Figure 3 micromachines-13-02212-f003:**
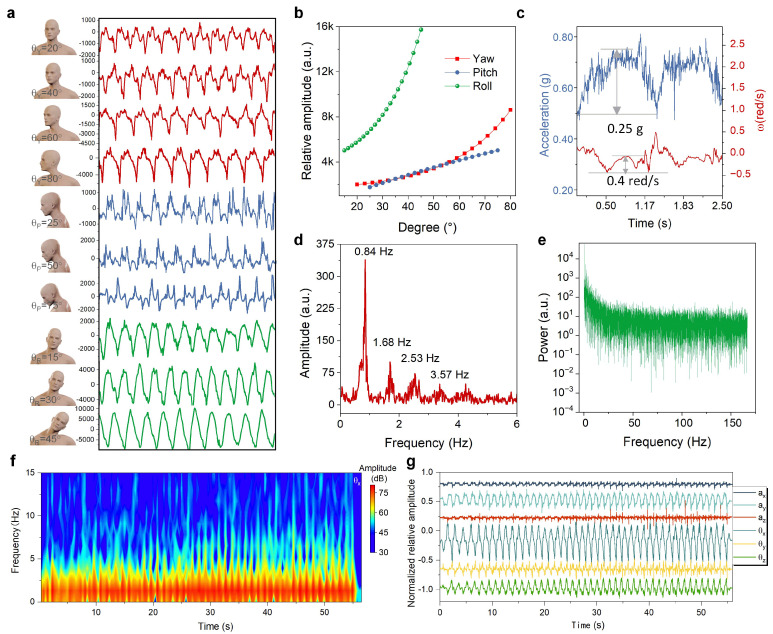
Inertial measurement results of head postures. (**a**) Continuous inertial angular velocity of the corresponding posture around the main axis. (**b**) Relative amplitude of head movements at different rotating angles. (**c**) The lowest resolution in the acceleration and angular velocity measurements when the pitch angle of the tested posture is 5°. (**d**,**e**) Frequency response of the amplitude and power spectrum of the head posture when θ_Yaw_ = 80°, respectively. (**f**) Short-time FT result of the head posture around the main axis when θ_Yaw_ = 80°. (**g**) Long-term inertial measured results of the head posture when θ_Yaw_ = 80°.

**Figure 4 micromachines-13-02212-f004:**
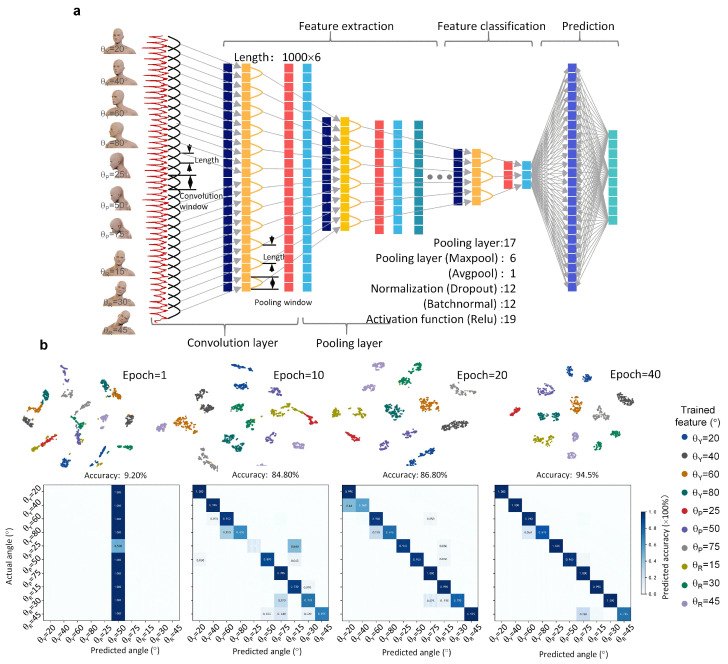
Machine learning for head posture recognition. (**a**) 2D CNN-based machine learning method. (**b**) Visualization cluster of the training via t-SNE (top) and the corresponding confusion matrix from the initial to the 40th epoch (bottom).

**Figure 5 micromachines-13-02212-f005:**
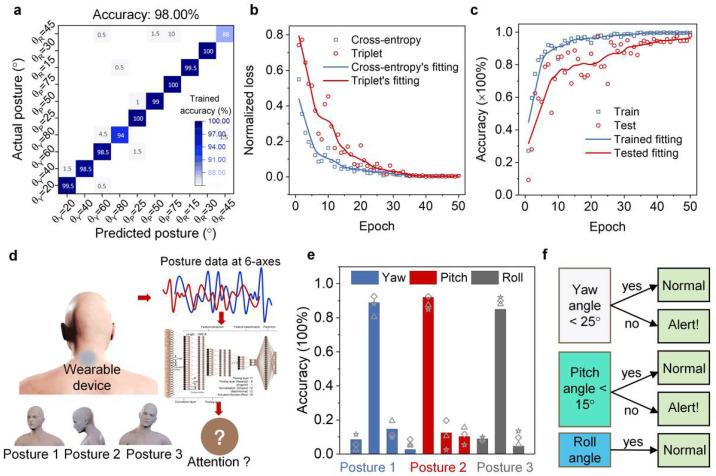
Applications of learning attention through head posture prediction. (**a**) Confusion matrix of real and predicted results under different head postures. (**b**) Trained and tested accuracy during deep learning. (**c**) Normalized loss results of the Ce and Tri function used for feature extractions and classifications. (**d**) Attention assessment block diagram for three new postures. (**e**) Attention-tested accuracy results of three different postures. (**f**) Presupposed attention assessment based on pose angles.

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
