# Peer review of "Attachable Inertial Device with Machine Learning toward Head Posture Monitoring in Attention Assessment"

_micromachines, 2022, doi:10.3390/mi13122212_

Round 1
Reviewer 1 Report
The authors proposed a fully-integrated attachable inertial device (AID) that comfortably monitors in-situ head posture at the neck and provides a machine learning-based assessment of attention.
There are some queries:
- Instead of 2D-like CNN, it should be 2D-CNN.
- There are typos.
- The authors may include related work based on attachable devices.
- The authors may include a flowchart of the complete system.
- The proposed processes should be revised in a more formal pseudocode template.
- Any comparison over other recent methods?
- Section 3 should be part of the method.
- Results and discussion need to be elaborately explained.
- The related work could be extended and incorporates more comprehensive discussions. Some discussions and important references are missing and should be added to the related work. E.g., "Gesture recognition from RGB images using CNN-attention based system", "A Convolution Neural Network and Classical Moments Based Feature Fusion model for Gesture Recognition", and "CNN Based Feature Extraction and Classification for Sign Language".
Author Response
We deeply appreciate reviewer’s positive comments to our works. According to the reviewer’s insightful comments, we have tried our best to address the comments by performing further experiments/analysis and revising our manuscript accordingly to meet with the requirements. In this response letter, the blue sections are the authors’ replies, and the red sections are the revised text in the manuscript, the figure numbers as well have been rearranged in order of the content. Additionally, we also invited two additional authors to help us improve the manuscript language and revision. The response is attached in the appendix.

Reviewer 2 Report
Thank you for your submissions. As a many-year researcher in inertial sensors, posture/activity recognition, signal processing, and machine learning for many years, I enjoyed the novelty of this article. The text is smoothly presented.
I approach this article from two separate perspectives:
— First, from a device design (hardware) perspective, I think the article's presentation is apparent, reasonable, and informative. I have no problem with this.
— Second, from the ML model (software) perspective, I think the article needs to be improved:
1. The confusion matrix is strange. I don't understand why the other FN/FPs were not drawn?
2. How is the universal accuracy calculated? Macro average? Is the number of recorded data balanced for each class?
3. One of the most significant issues is the comparison to the state-of-the-art methods, as in table 1. Each method has its own dataset, sensing technique, model approaches, and parameter configurations. Purely comparing the best accuracies of each is exceedingly problematic. Imagine I set up a dataset containing two classes—head nod, head shake—using ACC, Gyro, and video data, and then used a simple ML method that achieved 99% accuracy. Can I declare that my method is better than yours? Such a comparison is meaningless.
[32] is an angle estimation method. What is correct is to apply your dataset on their method and get a result by a detailed parameter tuning. Then apply its dataset on your method and get a result. That's the means to show the superiority of your method. Or, experiment with other publicly available datasets.
However, I think the purpose of your article is not to highlight that you have found some big advantage of machine learning for HP recognition, but to practice a kind of machine learning on the device you have developed. So, I think the article title and the tabled comparison are a bit out of focus. That's a trade-off you need to weigh.
4. The article mentions "real-time" several times in the introductory section, and the potential use of your device is bound to be primarily real-time. However, the article only examines the offline model. Although it is not necessarily required to demonstrate and outlook its real-time nature here, if this offline model requires long computation time and cost, it will definitely not be suitable for online real-time systems and will make the future use of your device much less promising. Since the title also highlights "Machine Learning", and the introduction of your device mentions its real-time application scenarios (e.g., online classes), it is worth analyzing the complexity and efficiency of the ML recognition method. Is there a potential application value for real-time? Or is it just a well-trained offline model that has a long way to go towards real-time applications? Putting a good posture/activity recognition system into real-time could refer to and cite: https://doi.org/10.1109/PerComWorkshops53856.2022.9767207 and https://doi.org/10.5220/0007398800470055.
An application scenario oriented recognition system research process can provide more ideas and references for the design of hardware devices and ML models. Here you may refer to the recently proposed "A Practical Wearable Sensor-Based Human Activity Recognition Research Pipeline".
5. When researching involving human physiological data, please be careful about data privacy, permission, and other academic ethical issues. I did not see these in any part of the article or in the appendix.
Minors:
1. Uniformity in mathematical accuracy. For example, if you use decimal points for percentages, 79% or 98% is abrupt/uncritical and should be 79.0% or 98.0%. It is recommended to use two decimal points (XX.XX%) in accordance with scientific convention.
2. There are quite a few problems with the figures. For example, some content is obscured (e.g., Figure 6e); generally, the layout is too tight with a poor split.
3. There are some glitches and typos in the text, which I highlighted in the PDF.
4. The inertial measurements are richly described. But what is an inertial sensor itself? What does it mean to the overall biosignal research, posture/activity recognition field, and a wide range of wearable-based applications? You need to "educate" the reader, especially in a journal article. You can refer to and cite one of the most detailed background introduction of the inertial sensor application in the relevant research domains: "Biosignal processing and activity modeling for multimodal human activity recognition."
Author Response

(The authors gave the same response as above.)

Round 2
Reviewer 1 Report
The authors addressed the reviewer's comments.
Reviewer 2 Report
The authors recognized my major opinion that the overemphasis on the accuracy of their ML model and comparisons with other state-of-the-art methods are both scientifically incorrect and takes away from the real innovative point the article is trying to make. This is reassuring.
The signing of the subject consent is briefly described. Right.
I deeply acknowledge the effort put by the authors to address all the academic issues I mentioned, and the comprehensive revision of the textual expression.
One small error need to be fixed: The newly added citation [23] has the wrong article title (should be without the "In" at the beginning).